# Metformin Protects the Intestinal Barrier by Activating Goblet Cell Maturation and Epithelial Proliferation in Radiation-Induced Enteropathy

**DOI:** 10.3390/ijms23115929

**Published:** 2022-05-25

**Authors:** Hyosun Jang, Soyeon Kim, Hyewon Kim, Su Hyun Oh, Seo Young Kwak, Hyun-Woo Joo, Seung Bum Lee, Won Il Jang, Sunhoo Park, Sehwan Shim

**Affiliations:** Laboratory of Radiation Exposure & Therapeutics, National Radiation Emergency Medical Center, Korea Institute of Radiological and Medical Science, Seoul 01812, Korea; sykim34@kirams.re.kr (S.K.); hw0227@kirams.re.kr (H.K.); sh9393@kirams.re.kr (S.H.O.); ksy@kirams.re.kr (S.Y.K.); joodengs@nate.com (H.-W.J.); sblee@kirams.re.kr (S.B.L.); zzang11@kirams.re.kr (W.I.J.); sunhoo@kirams.re.kr (S.P.)

**Keywords:** radiation-induced enteropathy, metformin, intestinal stem cells, goblet cells, intestinal barrier

## Abstract

Radiotherapy or accidental exposure to high-dose radiation can cause severe damage to healthy organs. The gastrointestinal (GI) tract is a radiation-sensitive organ of the body. The intestinal barrier is the first line of defense in the GI tract, and consists of mucus secreted by goblet cells and a monolayer of epithelium. Intestinal stem cells (ISCs) help in barrier maintenance and intestinal function after injury by regulating efficient regeneration of the epithelium. The Wnt/β-catenin pathway plays a critical role in maintaining the intestinal epithelium and regulates ISC self-renewal. Metformin is the most widely used antidiabetic drug in clinical practice, and its anti-inflammatory, antioxidative, and antiapoptotic effects have also been widely studied. In this study, we investigated whether metformin alleviated radiation-induced enteropathy by focusing on its role in protecting the epithelial barrier. We found that metformin alleviated radiation-induced enteropathy, with increased villi length and crypt numbers, and restored the intestinal barrier function in the irradiated intestine. In a radiation-induced enteropathy mouse model, metformin treatment increased tight-junction expression in the epithelium and inhibited bacterial translocation to mesenteric lymph nodes. Metformin increased the number of ISCs from radiation toxicity and enhanced epithelial repair by activating Wnt/β-catenin signaling. These data suggested that metformin may be a potential therapeutic agent for radiation-induced enteropathy.

## 1. Introduction

The gastrointestinal (GI) tract is a radiation-sensitive organ. Radiation-induced GI damage presents a major limitation of radiotherapy of abdominal and pelvic cancers. It is predicted to be a compulsory factor of survival in the event of nuclear accidents or radiological terrorism [1,2]. GI damage in acute radiation syndrome is characterized by nausea, diarrhea, and endotoxemia, and fibrotic changes with chronic inflammation can induce delayed radiation damage [3,4]. GI syndrome occurs at radiation doses higher than that required for inducing the hematopoietic syndrome, and is associated with the significant loss of crypts, ISCs, and other cell types [1,5,6]. Although several countermeasures already have been granted investigational new drug status (CBLB502, 5-AED, Bio300, EX-Rad) by the US Food and Drug Administration (FDA) and other promising protectors for hematopoietic damage are in development, there are still no FDA-approved effective mitigators for radiation-induced intestinal injury at present.

The intestinal barrier is the first line of defense in the GI tract, and mainly consists of a chemical and a physical barrier. The chemical barrier is mucus secreted by goblet cells, and the physical barrier includes a monolayer of epithelial cells and tight junctions (TJs) located at the cell membrane [7]. Barrier damage leads to increased epithelial permeability and bacterial translocation to internal organs, followed by leukocyte accumulation, cytokine and chemokine release, and inflammation [8,9].

Goblet cells play an essential role in maintaining and protecting the intestinal barrier by secreting mucins [10]. The mucus layer contains high concentrations of antimicrobial molecules and acts as a chemical barrier against the bacteria and pathogens residing in the intestinal lumen. In particular, mucin 2 (MUC2), the key component of mucus, is an important component of the intestinal mucus barrier, and is a secreted gel-forming mucin [11,12]. MUC2-deficient mice exhibit epithelial barrier dysfunction, dysbiosis, spontaneous intestinal inflammation, and tumorigenesis [13]. In patients with inflammatory bowel disease (IBD), goblet cell depletion is present in the colon in inflamed lesions compared to healthy lesions, and altered mucus layer thickness with reduced MUC2 has also been found [14,15]. However, research on goblet cell changes in the irradiated small intestine is limited.

Epithelial TJs are physical components of barrier integrity that maintain intestinal homeostasis by controlling paracellular permeability [16]. Intestinal epithelial TJs are highly sensitive to radiation exposure, and rapid disruption of TJ components decreases epithelial integrity in the irradiated intestine [17]. Clinical studies have shown that patients receiving radiotherapy have increased intestinal permeability and TJ disruption [18].

The GI tract has a rapid self-renewal rate of approximately 3–5 days. Intestinal stem cells help in barrier maintenance and intestinal function after injury by regulating efficient regeneration of the epithelium [19]. The proliferation and maintenance of intestinal stem cells is controlled by the Wnt/β-catenin signaling pathway [20]. Wnt pathway activation reinforces epithelial cell proliferation and reduces GI injury in mice.

Metformin is the most widely used antidiabetic drug in clinical practice. In addition to its hypoglycemic effects, metformin possesses various biological effects, including antioxidative, anti-inflammatory, and antiapoptotic properties [21,22,23]. Moreover, clinical data have shown that metformin usage is positively correlated with the overall survival of patients with liver cancer after radiotherapy [24]. Zhang et al. reported that metformin activated the Wnt/β-catenin signaling pathway and reduced neuronal injury in an animal model [25]. However, the effects of metformin on Wnt/β-catenin pathway activation in radiation-induced enteropathy have not been investigated. This study aimed to investigate whether metformin alleviated radiation-induced enteropathy by focusing on its role in maintaining intestinal barrier function.

## 2. Results

### 2.1. Metformin Alleviated Radiation-Induced Enteropathy and Partially Inhibited Inflammation Response in a Mouse Model

In order to investigate the effects of metformin on radiation-induced enteropathy, we exposed the abdomen of mice to 13.5 Gy of radiation using X-rad-320 and treated them with metformin (Figure 1A). We monitored the bodyweight loss in irradiated male C57BL/6 mice treated with or without metformin. Irradiated (IR) mice showed severe bodyweight loss. There was no significant difference between the metformin-treated IR group and the untreated IR group (Figure 1B). Furthermore, we analyzed the histopathological alterations in the irradiated small intestinal tissue. Crypt and villus destruction, epithelial detachment, inflammatory cell infiltration in the mucosa and submucosa, and thickness of the submucosa were shown in the irradiated intestine on day 6 (Figure 1C). Villus length and crypt numbers in the ileum were significantly increased in metformin-treated IR mice compared with those in the IR group (Figure 1E,F). Histological scores, including epithelial and crypt destruction, vascular dilation, and inflammation, were significantly reduced in the metformin-treated IR group compared to those in the IR group (Figure 1G).

Radiation-induced enteropathy is characterized by an inflammatory response with increased leukocyte accumulation and expression of (pro)inflammatory cytokines such as interleukin (IL)-1β and matrix metalloproteinase 9 (MMP9). We examined the effects of metformin on the inflammatory responses in irradiated intestines. There was no significant difference in inflammatory scores between the IR group and metformin-treated IR group (Figure 1H). Increased levels of myeloperoxidase (MPO), a neutrophil marker, correspond to the severity of inflammation in intestinal disease. The number of Mpo-positive cells was markedly increased in the intestinal mucosa of IR mice compared to that of control mice (Figure 1D). Metformin treatment partially inhibited the recruitment of Mpo-positive cells in the irradiated intestine (Figure 1D). MMP9 and IL1β levels are known to increase during acute radiation-induced enteropathy, and play pivotal roles in the inflammatory response [26]. We observed a significant increase in the mRNA levels of Mmp9 and IL1β in irradiated intestinal tissues (Figure 1I). Metformin treatment did not significantly attenuate the mRNA expression of inflammatory cytokines compared to the untreated IR group (Figure 1I). Therefore, metformin alleviated radiation-induced enteropathy and partially inhibited inflammatory responses in a mouse model.

### 2.2. Metformin Protected Intestinal Barrier Function by Indirectly Regulating Goblet Cell Maturation in Radiation Damage

Bacterial translocation from the intestinal lumen to other organs indicates damage to the epithelial barrier in the GI tract. We assessed bacterial translocation in the mesenteric lymph nodes of a radiation-induced enteropathy mouse model to determine whether metformin affected the intestinal epithelial barrier after irradiation. The percentage of positive colonies in the mesenteric lymph nodes was markedly higher in the IR group than in the control group (Figure 2A). The percentage of bacterial translocation decreased in metformin-treated IR mice compared with that in untreated IR mice (Figure 2A). Mucus secreted by goblet cells is a critical chemical barrier of the intestine that protects against the invasion of antigens and pathogens. Disruption of the mucus layer results in increased intestinal permeability, followed by bacterial translocation and systemic inflammatory responses. As shown in Figure 2B,C, the IR group showed a significant decrease in Alcian-blue- and PAS-stained goblet cells, consistent with the decreased expression of acidic and neutral mucins, unlike the control group (Figure 2B,C). The metformin-treated IR groups showed increased expression of Alcian-blue-stained acidic and PAS-stained neutral mucins and higher numbers of goblet cells per villus in the small intestine compared to the IR group (Figure 2B,C).

Mucin 2 (MUC2) is a differentiated goblet cell marker that is a critical component of the intestinal mucous barrier [12]. Muc2-positive cells showed malfunctioning enlargement, and their numbers decreased in the IR group (Figure 2D). Metformin treatment alleviated radiation-exposure-induced damage to goblet cells (Figure 2D). In addition, the mRNA levels of Muc2 and trefoil factor 3 (Tff3), another goblet cell differentiation marker, were decreased in the IR group compared to their levels in the control group and significantly increased in the metformin-treated IR group (Figure 2E). Atonal BHLH transcription factor 1 (ATOH-1), growth factor independent 1 transcriptional repressor (GFI-1), and SAM pointed domain-containing ETS transcription factor (SPDEF) are required for proper goblet cell maturation and mucin production [27]. We also found that metformin significantly increased the mRNA levels of Atoh-1, Gfi-1, and Spdef in the irradiated intestines (Figure 2F). To investigate the direct effects of metformin on the maturation of irradiated goblet cells, we performed in vitro experiments using HT29 cells. However, we did not find any direct effects of metformin on goblet cell maturation under radiation exposure conditions (data not shown). These results suggested that metformin protected against intestinal barrier damage by indirectly enhancing goblet cell maturation in irradiated mice.

### 2.3. Metformin Enhanced Tight-Junction Expression and Epithelial Proliferation in Irradiated Intestine

TJs, which are transmembrane proteins such as claudin 3 (CLDN3), occludin, and zonula occludens 1 (ZO1), create an epithelial barrier that regulates the paracellular movement of water and solutes across epithelia. CLDN3 and ZO1 are particularly sensitive to radiation exposure, and mainly regulate intestinal barrier function [28]. Cldn3 protein expression was upregulated in the irradiated intestine following metformin treatment (Figure 3A). In addition, we found that the mRNA levels of *Cldn3* and *Zo1* were markedly increased in the metformin-treated IR group compared to those in the untreated IR group (Figure 3D).

Increased epithelial cell proliferation contributes to enhancing intestinal epithelial barrier function [26,29]. Villin is an enterocyte marker involved in regulating epithelial integrity and apoptosis. Furthermore, villin protein expression was broadly localized in the irradiated intestine after metformin treatment (Figure 3B). Moreover, we found that the mRNA levels of *Villin* were significantly increased in the metformin-treated IR group compared to those in the untreated IR group (Figure 3D). Ki-67 is expressed by cells in an active state of proliferation, and it identifies regenerative crypts [30]. Ki-67-labeled intestinal epithelial cells decreased in the IR group, whereas they were increased in the metformin-treated IR mice (Figure 3C). These results suggested that metformin restored tight-junction expression and increased epithelial proliferation after radiation exposure.

### 2.4. Metformin Promoted Stem Cell Properties with Wnt/B-Catenin Pathway Activation in Radiation Damage

As the intestinal stem cell is the weak point for radiation exposure, damaged stem cells retarded epithelial cell proliferation and regeneration of the intestine. Additionally, the Wnt/β-catenin pathway in stem cells plays a critical role in self-renewal and proliferation. Subsequently, we performed immunohistochemistry for the intestinal stem cell marker Olfm4 to evaluate intestinal stem cell damage in metformin-treated irradiated mice. The intestines of the IR group exhibited a few Olfm4-stained cells in the crypts, whereas those of the metformin-treated IR group showed an increased number of Olfm4-positive cells in the crypts (Figure 4A). The mRNA levels of Olfm4 were significantly increased in metformin-treated IR mice compared to those in the IR group (Figure 4C). We analyzed the expression of other intestinal stem cell markers, such as leucine-rich repeat-containing G protein-coupled receptor 5 (LGR5) and SRY-box transcription factor 9 (SOX9). In irradiated intestines, the mRNA levels of *Lgr5* and *Sox9* were significantly lower than those in the intestines of control mice. Metformin treatment increased the mRNA expression of *Lgr5* and *Sox9* in irradiated intestines (Figure 4C). We also found that metformin increased the protein and mRNA expression of β-catenin in the irradiated crypts (Figure 4B,D). The mRNA levels of Wnt/b-catenin-related genes, such as *Myc* and *Ascl2*, were significantly increased in metformin-treated irradiated mice compared to those in the untreated IR group (Figure 4D). These results suggested that metformin activated the Wnt/β-catenin pathway and improved the intestinal stem cell properties following radiation-induced enteropathy.

### 2.5. Metformin Increased Budding Organoid Formation and Epithelial Proliferation by Activating Wnt/B-Catenin Signaling

As it is well known that metformin promotes cell proliferation during tissue repair, we explored the direct function of metformin in irradiated intestinal stem and epithelial cells. To further analyze the specific effects of metformin on the intestinal stem cell population, we developed a mouse-derived intestinal organoid culture system exposed to irradiation. Intestinal organoid formation and maintenance require several biological processes, including stem cell proliferation, self-renewal, differentiation, and new crypt formation [31]. Organoids exposed to irradiation showed significantly decreased numbers of budding crypts and total crypts compared with the non-IR organoid groups (Figure 5A,B). After irradiation, treatment with metformin (20 μM) rescued the organoids from radiation-induced damage and improved the budding crypt/total crypt ratio (Figure 5A). Since the distribution of MTT-stained organoids can be easily inspected by light microscopy, we performed these assays in metformin-treated irradiated organoids. MTT-stained organoids dramatically decreased under irradiated conditions, and metformin treatment increased MTT-stained survival in the irradiated organoids (Figure 5B). The MTT absorbance also significantly increased in the metformin-treated group compared to that in the IR group (Figure 5B).

To evaluate the direct effects of metformin on epithelial cells, we used the rat small intestinal epithelial cell line 6 (IEC-6), which shares beneficial characteristics with small intestinal epithelial transit-amplifying cells [32]. Treatment with metformin significantly increased proliferation compared to 2 Gy and 4 Gy irradiated IEC-6 cells (Figure 5C). We also found that metformin increased β-catenin expression in irradiated epithelial cells (Figure 5D). The mRNA levels of Wnt/b-catenin-related genes, such as *Myc*, *Ascl2*, and *Axin2*, were significantly increased in metformin-treated irradiated IEC-6 cells compared to those in the 2 Gy and 4 Gy IR groups (Figure 5E). These results suggested that metformin increased intestinal stem cell properties and epithelial proliferation by activating the Wnt/β-catenin pathway under irradiation.

## 3. Discussion

Accidental exposure to or radiotherapy using high-dose radiation can cause severe damage to healthy intestinal tissues of the GI tract [33]. Metformin is a widely used oral hypoglycemic drug for type 2 diabetes mellitus worldwide, and its multiple effects have been widely studied. In addition, metformin enhanced tumor response to radiotherapy in patients with cancer and diabetes mellitus, and improved survival in diabetic colorectal cancer patients [34,35]. Ahmadi et al. reported that metformin decreased intestinal permeability and inflammatory response in older obese mice by beneficially regulating the gut microbiota [36]. According to our data, although metformin treatment did not effectively inhibit the radiation-induced inflammation response, it enhanced intestinal [35] barrier function in irradiated intestines and alleviated radiation-induced enteropathy.

The intestinal barrier is the first line of defense in the GI tract, preventing mucosal tissue from being exposed to microbial antigens and pathogens. TJs play critical roles in intestinal homeostasis by regulating epithelial permeability and integrity [16]. Barrier damage with tight-junction loss is a well-known cause of radiation-induced enteropathy [18,28]. Chen et al. reported that adenosine-monophosphate-activated kinase activation by metformin accelerated tight-junction molecules, such as Zo-1 and occludin, and alleviated DSS-induced colitis [22]. In a radiation-induced enteropathy mouse model, we also found that metformin treatment increased tight-junction expression in irradiated intestines and inhibited bacterial translocation to mesenteric lymph nodes.

Goblet cells secrete mucus to maintain a healthy intestine [10]. The mucus layer plays a vital role in supporting lubrication to for digestion, participating in cell signaling pathways, and protecting the epithelium from invading antigens, toxins, and other environmental irritants. MUC2, a gel-forming mucin, is an essential component of the protective mucus layer in the intestine [11]. Mice with MUC2 depletion developed spontaneous colitis with weight loss, changes in stool consistency, and increased inflammatory cytokine expression [13]. In addition, these mice showed severe inflammation and disease activity in a DSS-induced colitis model [13]. Patients and animal studies have reported that several GI diseases, including IBD, GI cancer, and intestinal infections, cause considerable changes in mucin quality and quantity. Indeed, reduced production of mucosal-barrier-related molecules, such as antimicrobial peptides and MUC2, with goblet cell loss and barrier dysfunction was observed in the intestinal lesions of IBD patients [14,15,37]. However, there are limited reports on changes in goblet cells and MUC2 expression in radiation-induced small intestinal injury. In this study, we found that irradiation induced the malformation of goblet cells and decreased MUC2 expression in the small intestine.

The host-sensing gut microbes or their metabolites regulate mucus secretion from goblet cells, such as short-chain fatty acids and Th2 cytokines, including IL-5 and IL-13 [7]. In our study, metformin inhibited radiation-induced goblet cell damage and increased the expression of goblet cell maturation markers, including MUC2 and TFF3. Goblet cell generation and maturation-regulated genes (*Atoh-1, Gfi-1*, and *Spdef*) increased in metformin-treated IR mice. We performed in vitro experiments to determine the direct effects of metformin on goblet cell maturation. However, we confirmed that metformin did not directly affect goblet cell maturation under radiation exposure. Others have reported that metformin alleviated DSS-induced colitis by increasing the levels of potential probiotics (e.g., *Akkermansia muciniphila*) [38] and attenuated aging-related leaky gut by modulating the microbiome–goblet cell–mucin axis [36]. Therefore, metformin-induced goblet cell maturation in a mouse model may be related to microbiome alterations in the irradiated intestine.

A high dose of radiation principally leads to impairment of epithelial regeneration by eliminating intestinal stem cells (ISC) [39,40]. A dysfunctional epithelium significantly reduces barrier integrity and promotes the influx of bacterial pathogens, resulting in sepsis and death [39]. Therefore, Wnt/β-catenin signaling is involved in the maintenance of intestinal epithelial homeostasis and repair by regulating the self-renewal and proliferation of ISCs [40,41]. Wnt activation stimulates regeneration and mitigates lethality after ablative ionizing-radiation-induced intestinal injury [42]. In the intestinal crypts, our data showed that radiation-induced damage markedly reduced ISC markers, such as Olfm4, Lgr5, and Sox9. Metformin treatment upregulated ISCs in the irradiated intestine by activating the Wnt/β-catenin pathway. Altogether, metformin might activate ISC self-renewal and proliferation by activating the Wnt/β-catenin pathway, thereby enhancing barrier function in radiation-induced enteropathy.

Intestinal organoids are consistent with the crypt–villus structure, including ISCs, Paneth cells, enteroendocrine cells, and enterocytes [43]. Intestinal organoid growth depends on the presence of healthy intestinal stem cells [43]. Therefore, this ex vivo organoid system provides a good platform for analyzing and estimating the efficacy of any potential GI radio-mitigator and to mechanistically demonstrate the effects on ISC properties. Using an ex vivo model, we found that metformin increased organoids’ budding and survival rate under radiation conditions. In addition, metformin upregulated epithelial proliferation and increased the expression of Wnt/β-catenin-related genes in irradiated IEC-6 cells. Therefore, in an organoid model developed from the mouse intestinal epithelium and IEC-6 cell line, we demonstrated that metformin rescued ISCs from radiation toxicity and enhanced epithelial repair by activating Wnt-β-catenin signaling.

In summary, we found that metformin regulated goblet cell maturation and epithelial permeability in radiation-induced intestinal damage. In addition, metformin increased epithelial proliferation, activated the Wnt/β-catenin pathway, and reinforced the intestinal epithelial barrier. These results indicated that metformin protected the intestinal barrier by modulating goblet cell maturation and ISC properties in radiation-induced epithelial damage. This suggested that metformin may be used as a powerful therapeutic agent for radiation-induced enteropathy.

## 4. Material and Methods

### 4.1. Mice

Seven-week-old male C57BL/6 mice were obtained from DooYeol Biotech (Seocho-gu, Seoul, Korea) and maintained under specific pathogen-free conditions at the animal facility of the Korea Institute of Radiological and Medical Sciences (KIRAMS). All mice were housed in a temperature-controlled room with a 12 h light/dark cycle, and food and water were provided ad libitum. The mice were acclimated for 1 week and were randomly divided into the following groups: (1) control, (2) IR, and (3) IR + Met. All animal experiments were approved and performed in accordance with the guidelines of the Institutional Animal Care and Use Committee of the KIRAMS.

### 4.2. Irradiation and Treatment of Metformin

The mice were anesthetized with a combination treatment of alfaxalone (85 mg/kg; Alfaxan^®^; Careside, Gyeonggi-do, Korea) and xylazine (10 mg/kg; Rompun^®^; Bayer Korea, Seoul, Korea). They were irradiated with a single dose of 13.5 Gy (2 Gy/min) to the abdomen using an X-RAD 320 X-ray irradiator (Softex, Goyang-si, Goyang-si, Korea). After radiation exposure, the mice were orally treated with 500 mg/kg/day of metformin (Diabex, Daewoong Pharm. Co., Ltd., Seoul, Korea) for 6 days.

### 4.3. Histological Analysis of the Small Intestine

Intestinal tissues of metformin-treated mice were fixed with 10% neutral buffered formalin solution, embedded in paraffin wax, and sectioned transversely at a thickness of 4 µm. The sections were then stained with H&E. The histological score and inflammation score were quantified in H&E-stained sections of the ileum as a reference [28,44]. The severity of radiation-induced enteropathy was assessed based on the degree of maintenance of the epithelial layers, crypt destruction, vascular dilation, and infiltration of inflammatory cells in the lamina propria. For the immunohistochemical analysis, the sections were treated with antigen retrieval and 0.3% hydrogen peroxide in methyl alcohol. After three washes with PBS, the sections were incubated with blocking solution (Vector ABC Elite kit; Vector Laboratories, Burlingame, CA, USA) and then incubated with anti-Mpo (Abcam, Cambridge, UK), Muc2, Cldn3 (Invitrogen, Carlsbad, CA, USA), Ki-67 (Acris), Olfm4 (Invitrogen), and-catenin (Cell Signaling Technology, Danvers, MA, USA) antibodies. After three washes in PBS, the sections were incubated with a horseradish-peroxidase-conjugated secondary antibody (Dako, Carpinteria, CA, USA). The peroxidase reaction was developed using diaminobenzidine substrate (Dako) prepared according to the manufacturer’s instructions, and the slides were counterstained with hematoxylin. In order to identify goblet cell alterations, sections were stained with Alcian blue and periodic acid–Schiff (PAS). The Alcian blue stained acidic mucopolysaccharides, whereas the PAS stained neutral mucosubstances.

### 4.4. RNA Extraction, Reverse Transcription Polymerase Chain Reaction (RT-PCR), and Real-Time PCR Quantification

The collected intestinal tissues were immediately snap-frozen and stored at −80 °C until RNA extraction was performed. Total RNA was isolated from whole intestine and IEC-6 cells using TRIzol reagent (Invitrogen, Carlsbad, CA, USA). The cDNA was produced using the AccuPower RT premix (Bioneer, Daejeon, Korea) according to the manufacturer’s protocol. The real-time RT-PCR was performed using a LightCycler 480 system (Roche, San Francisco, CA, USA). The primer sequences are listed in Table 1. The expression levels of each target gene were normalized to those of GAPDH. Cycle threshold values were used to validate the relative mRNA levels using the 2^−∆∆Ct^ method.

### 4.5. Bacterial Translocation Assay

To analyze bacterial translocation from the intestinal lumen to the mesenteric lymph nodes, we collected the mesenteric lymph nodes from the control, IR, and IR + Met-treated mice under sterile conditions. The supernatant of the homogenated lymph node was spread onto MacConkey agar (BD Biosciences, Palo Alto, CA, USA) and incubated at 37 °C overnight.

### 4.6. Culture, Budding, and MTT Assay of Mouse Intestinal Organoids

The crypts were isolated from the small intestines of mice, and isolated crypts were cultured as previously reported [45]. The organoids were cultured onto four-well plates at the same ratio, and then exposed to 137Cs γ-rays from a Gamma Cell-3000 irradiator (MDS Nordion International, Kanata, Ontario, Canada). Metformin (20 μM; Sigma-Aldrich, St. Louis, MO, USA) was administered after irradiation on day 0, and the medium was changed every three days. For examination of intestinal organoid viability after irradiation, we performed methylthiazolyl tertrazolium (MTT) assays, as previously reported [46]. Briefly, organoids were incubated with 10% MTT for 2–3 h, and the insoluble formazan crystal were dissolved in solubilization solution. MTT-stained organoids indicated surviving cells.

### 4.7. Cell Culture, Irradiation, and Viability Assay

IEC-6 cells were grown in Dulbecco’s Modified Eagle Medium (Gibco, Grand Island, NY, USA) supplemented with 10% fetal bovine serum (Gibco) and 1% penicillin/streptomycin at 37 °C with 5% CO_2_. The cells were exposed to 2 Gy or 4 Gy γ-rays with a 137Cs γ-ray source (Atomic Energy of Canada, Chalk River, ON, Canada) at a dose rate of 3.81 Gy/min and treated with metformin (20 μM; Sigma-Aldrich). After 48 h of incubation, the Cell Counting Kit-8 reagent was added into the well of IEC-6 cells, and the absorbance at 450 nm was measured.

### 4.8. Statistical Analysis

Data were analyzed using GraphPad Prism software. Data are presented as the mean ± standard error of the mean values in mouse experiments or the mean ± standard deviation of the mean values for in vitro experiments. The statistical significance of differences was evaluated using Student’s t-test or one-way analysis of variance with Tukey’s multiple comparison test. Statistical significance was set at *p* < 0.05.

## Figures and Tables

**Figure 1 ijms-23-05929-f001:**
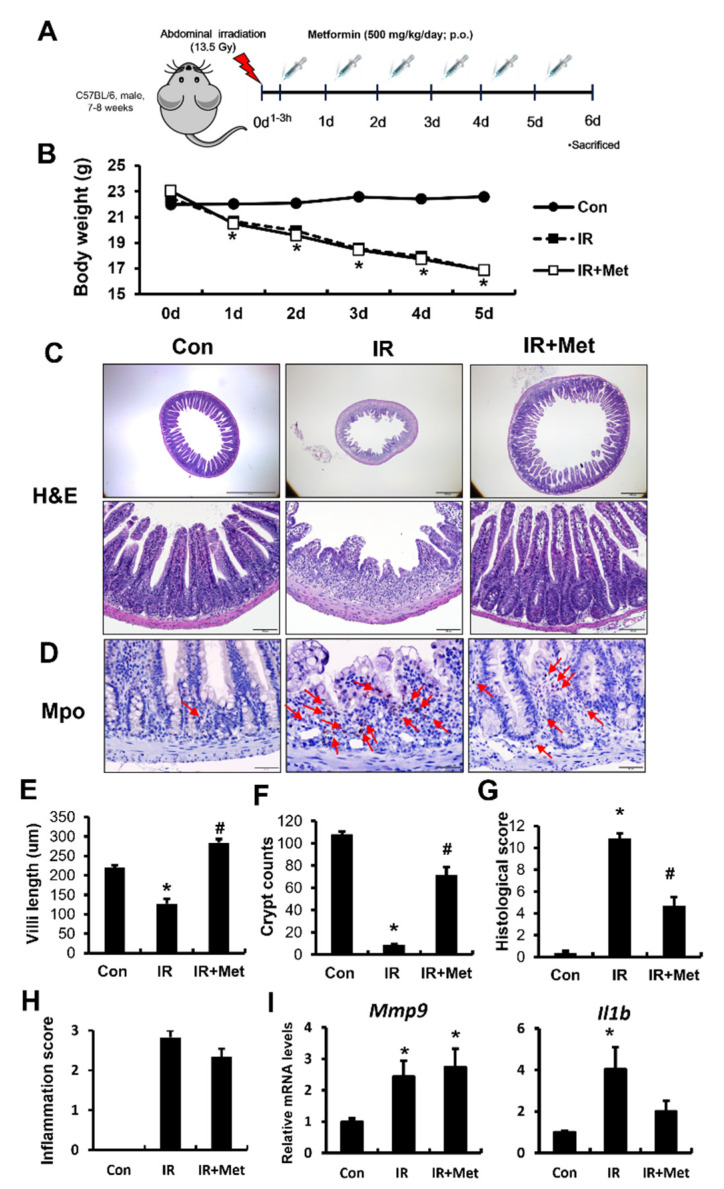
Metformin alleviated radiation-induced enteropathy and partially inhibited inflammation response in a mouse model. (**A**) Experimental schematic; (**B**) bodyweight; (**C**) hematoxylin–eosin (H&E) stain; (**D**) immunohistochemistry of myeloperoxidase (Mpo), a neutrophil marker; (**E**) villi length; (**F**) the number of crypt cells in the small intestine of control (Con), irradiated (IR), and metformin-treated IR (IR + Met) mice at 6 days. Bar = 50 μm; (**G**) histological score assessed by the degree of maintenance of epithelial structure, crypt destruction, vascular dilation, and inflammatory-cell infiltration in the mucosa (0 = none, 1 = mild, 2 = moderate, 3 = high); (**H**) inflammation score; and (**I**) mRNA levels of matrix metallopeptidase 9 (Mmp9) and interleukin (IL)-1β in the intestines from the Con, IR, and IR + Met groups. The red arrow indicates Mpo-positive neutrophils. Data are presented as the mean ± standard error of the mean; *n* = 6 mice per group. * *p* < 0.05 compared to the Con group; # *p* < 0.05 compared to the IR group.

**Figure 2 ijms-23-05929-f002:**
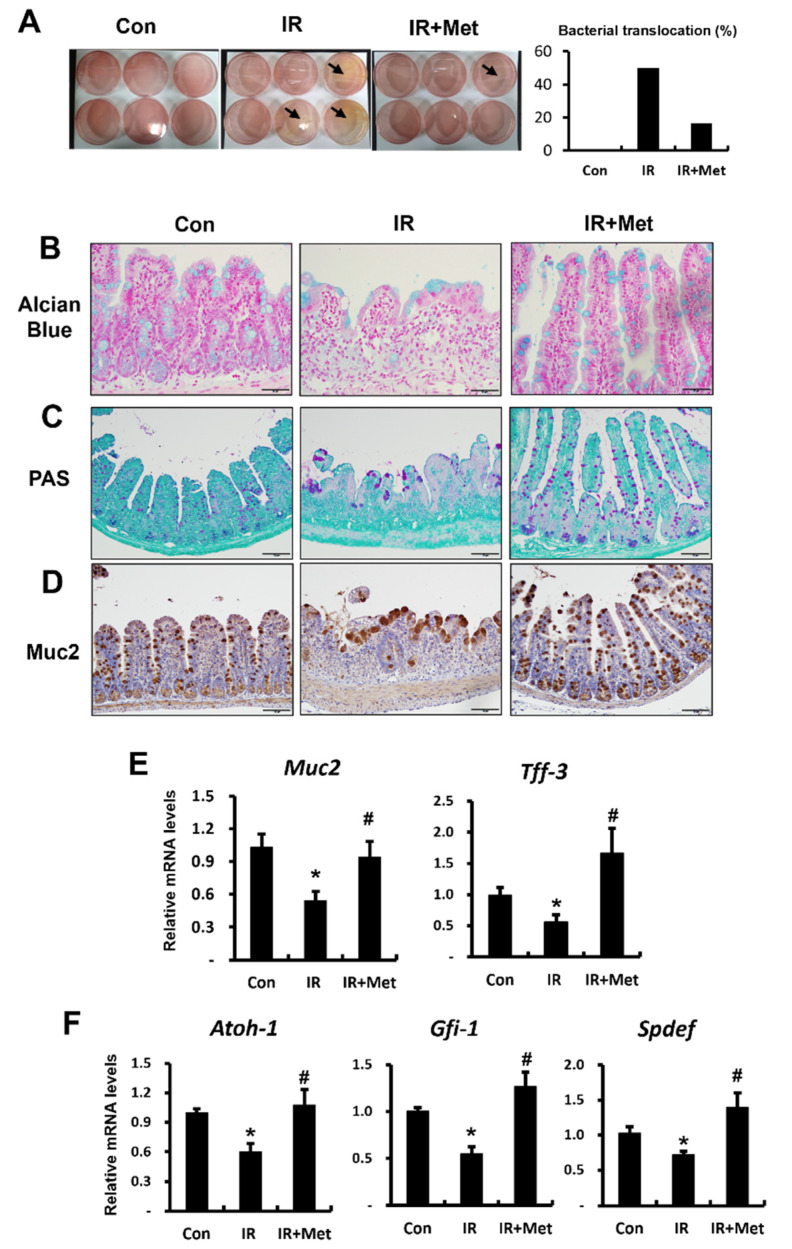
Metformin protected intestinal barrier function by indirectly regulating goblet cell maturation in radiation damage. (**A**) The percentage of bacterial translocation on mesenteric lymph nodes; (**B**) Alcian blue stain and (**C**) periodic acid–Schiff (PAS) stain for acidic and neutral mucin statin in the goblet cells; (**D**) immunohistochemistry of Mucin2 (Muc2) in the small intestine of control (Con), irradiated (IR), and metformin-treated IR (IR + Met) mice; (**E**) mRNA levels of Muc2, trefoil factor 3 (Tff3) of the intestinal tissue; (**F**) mRNA levels of atonal BHLH transcription factor 1 (Atoh-1), growth factor independent 1 transcriptional repressor (Gfi-1), and SAM pointed domain containing ETS transcription factor (Spdef) in the ileum of Con, IR, and IR + Met mice. Bar = 20 μm. Data are presented as the mean ± standard error of the mean; *n* = 6 mice per group. * *p* < 0.05 compared to the Con group; # *p* < 0.05 compared to the IR group.

**Figure 3 ijms-23-05929-f003:**
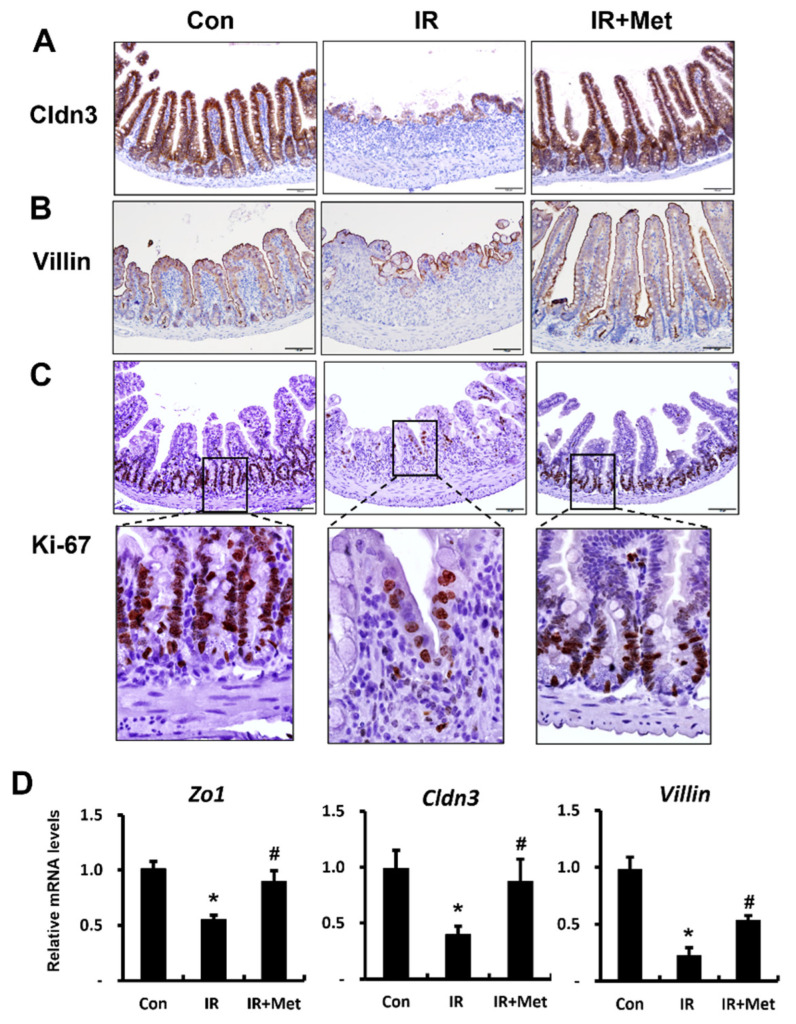
Metformin enhanced tight-junction expression with epithelial proliferation in the irradiated intestine. Immunohistochemistry of (**A**) Claudin3 (Cldn3), (**B**) Villin, and (**C**) Ki-67 in the intestine of control (Con), irradiated (IR), and metformin-treated IR (IR + Met) mice at 6 days after radiation exposure. Bar = 50 μm. (**D**) The mRNA levels of Zonula occludens1 (*Zo1), Cldn3, and Villin* in the Con, IR, and IR + Met groups, as determined by real-time RT-PCR. Levels of mRNA expression are shown as the fold induction relative to the expression in Con group. Data are presented as the mean ± standard error of the mean; *n* = 6 mice per group. * *p* < 0.05 compared to the Con group; # *p* < 0.05 compared to the IR group.

**Figure 4 ijms-23-05929-f004:**
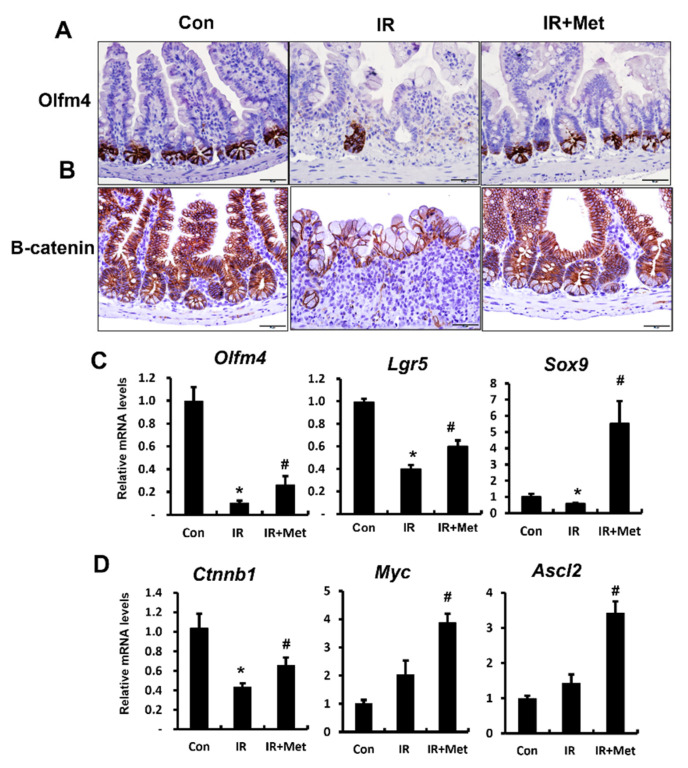
Metformin promoted stem cell properties with Wnt/b-catenin pathway activation in radiation damage. Immunohistochemistry of (**A**) Olfactomedin4 (Olfm4) and (**B**) β-catenin in the small intestine of control (Con), irradiated (IR), and metformin-treated IR (IR + Met) mice. Bar = 50 μm. (**C**) The mRNA levels of stem cell property markers, such as *Olfm4,* leucine-rich repeat containing G protein-coupled receptor 5 *(Lgr5)*, and SRY-box transcription factor 9 *(Sox9)* in Con, IR, and IR + Met groups. (**D**) The mRNA levels of β-catenin *(**Ctnnb1)* and Wnt/b-catenin-related genes *(Myc* and Achaete-scute complex homolog 2; *Ascl2*) in the Con, IR, and IR + Met groups, as determined by real-time RT-PCR. Levels of mRNA expression are shown as the fold induction relative to the expression in Con group. Data are presented as the mean ± standard error of the mean; *n* = 6 mice per group. * *p* < 0.05 compared to the Con group; # *p* < 0.05 compared to the IR group.

**Figure 5 ijms-23-05929-f005:**
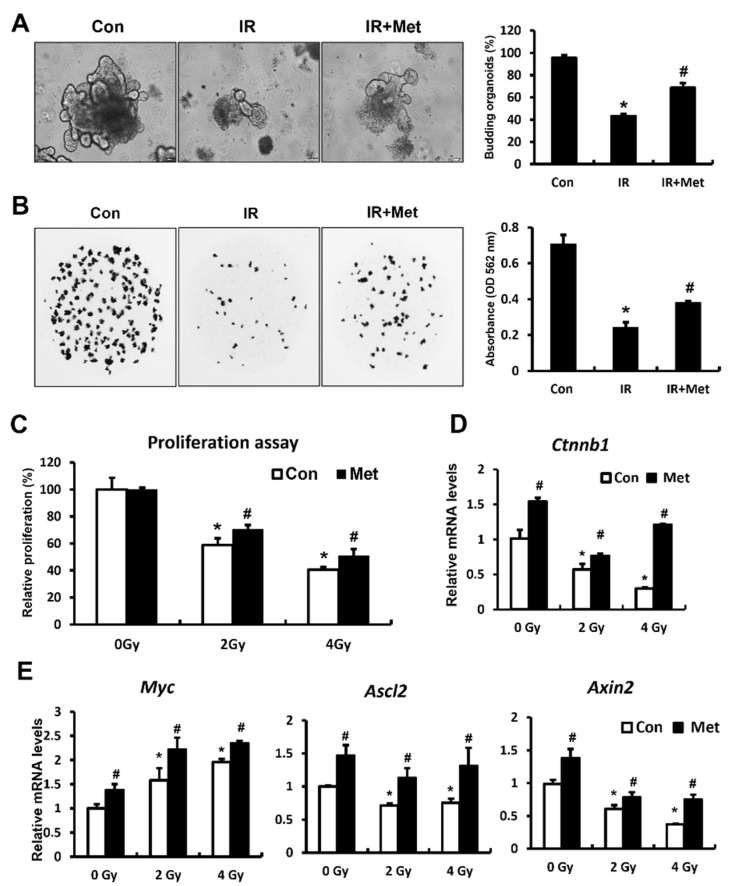
Metformin increased intestinal organoid formation and epithelial proliferation by regulating Wnt/b-catenin activation. (**A**) Morphology and percentage of budding organoid in nonirradiated (Con), irradiated (IR), and metformin-treated IR (IR + Met) intestinal organoid groups; (**B**) MTT assay in metformin-treated IR organoids; (**C**) proliferation assay in the irradiated IEC-6 cells-treated with metformin. mRNA levels of (**D**) β-catenin *(**Ctnnb1)* and (**E**) Wnt/b-catenin-related genes *(Myc, Ascl2,* and *Axin2)* in the Con, IR, and IR + Met groups. Levels of mRNA expression are shown as the fold induction relative to the expression in Con group. Data are presented as the mean ± standard error of the mean; *n* = 6 mice per group. * *p* < 0.05 compared to the Con group; # *p* < 0.05 compared to the IR group.

**Table 1 ijms-23-05929-t001:** Primer sequences used in experiments.

Species	Primer	Forward (5′–3′)	Reverse (5′–3′)
Mouse	*Mmp9*	GCCCTGGAACTCACACGACA	TTGGAAACTCACACGCCAGAAG
	*Il1β*	GCAACTGTTCCTGAACTCA	CTCGGAGCCTGTAGTGCAG
	*Muc2*	GCTGACGAGTGGTTGGTGAATG	GATGAGGTGGCAGACAGGAGAC
	*Tff-3*	TGGTCCAAGGGTAGCAAGCAT	CAGCCTTGTGTTGGCTGTGAG
	*Atoh-1*	GTGGGGTTGTAGTGGACGAG	GTTGCTCTCCGACATTGGG
	*Gfi-1*	AGAAGGCGCACAGCTATCAC	GGCTCCATTTTCGACTCGC
	*Spedf*	GGAGAAGGCAGCATCAGGA	CCAGGGTCTGCTGTGATGT
	*Zo1*	AGGACACCAAAGCATGTGAG	GGCATTCCTGCTGGTTACA
	*Cldn 3*	AAGCCGAATGGACAAAGAA	CTGGCAAGTAGCTGCAGTG
	*Villin*	CACCTTTGGAAGCTTCTTCG	CTCTCGTTGCCTTGAACCTC
	*Olfm4*	GCTGGAAGTGAAGGAGATGC	ACAGAAGGAGCGCTGATGTT
	*Lgr5*	TCAGTCAGCTGCTCCCGAAT	CGTTTCCCGCAAGACGTAAC
	*Sox-9*	CTCGCATACCTCCCTTCC	TTCCAGCAGTCACTAGGC
	*b-catenin*	ACTGCTGGGACTCTG	TGATGGCGTAGAACAG
	*Myc*	GGGACAGTGTTCTCTGCCTCTGC	AAGTTGCCACCGCCACCGTCATC
	*Ascl2*	GCCTACTCGTCGGAGGAA	CCAACTGGAAAAGTCAAGCA
	*Gapdh*	AAGATGGTGATGGGCTTCCCG	TGGCAAAGTGGAGATTGTTGCC
Rat	*b-catenin*	TCCCAATCAGTTGGTCAGCC	GAGAAAATCGCCCTTGAGGC
	*Myc*	GTGCTGCATGAAGAGACACC	CAACCTCTGGCTTCCTACCC
	*Ascl2*	TCTCTGTCCTGCACCTCTACATCC	AGCTGCTGTCCTCCGACGAG
	*Axin2*	CAGGACCCACATCCTTCT	ACGCGGAGGTGCACGCGG
	*Gapdh*	GAAGGTGAAGGTCGGAGTC	GAAGATGGTGATGGGATTTC

## Data Availability

Not applicable.

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
