# Peer review of "Metformin Protects the Intestinal Barrier by Activating Goblet Cell Maturation and Epithelial Proliferation in Radiation-Induced Enteropathy"

_ijms, 2022, doi:10.3390/ijms23115929_

Round 1
Reviewer 1 Report
The authors have reported on the effects of Metformin, a drug with multiple biologic activities and used in the treatment of Diabetes, to radiation protection with respect to enteropathy and the inflammatory response in the intestine. The authors look at multiple parameters of biology, then test them after irradiation. The studies are of interest and are provocative. I have given the authors some suggestions for future studies, and how they can amplify the current observations to a more mechanistic approach to understand the effects of Metformin.
Specific Comments:
Abstract:
The abstract should contain some quantitative information: villus length, number of intestinal stem cells, or other measures of goblet or intestinal barrier function.
Introduction:
The authors should summarize many of the other radiation protectors and mitigators that have been reported with respect to the gastrointestinal syndrome. Some reference to the animal models that have been used in addition to the mouse. Radiation doses that induce the GI Syndrome. Some explanation of the difference between hematopoietic syndrome and GI syndrome should be included.
Results:
The results are largely restricted to quantitative information, such as in Figure 1, which looks at body weight, H & E stain of intestine, histochemistry and myeloperoxidase activity is a neutrophil marker for inflammation, and villus length. This is largely quantitative and histopathology. It would help to have a dose response curve of various doses of irradiation and various times of measurement. Pathologic sections were taken at one time point. The survival curve of animals with each of several different radiation doses should be shown. Body weight is helpful, but the survival would be of interest. Someone should mention the mouse strain and whether males or females were used.
The authors use mucin2 (Muc2) as a marker of differentiated goblet cells. In Figure 2, they show goblet cells as a measure of intestinal barrier function. However, barrier function is usually quantitated by either fluorescent beads gavaged into the intestine, and then measured as leaking into the peripheral blood, or other measures of integrity of tight junctions between epithelial cells such as occludins, or i-CAM, or other adhesion molecules. However, these results are interesting. The authors look at bacterial translocation in panel A. They use Alcian Blue and PAS staining, as well as, Muc2 staining. In the Muc2 staining, the magnification factors are different and significant damage in the irradiation panel that is shown in the middle that is not seen in either the IR+Met or Control. A dose response curve of irradiation and a dose response curve of Metformin should be shown. Metformin is often given in drinking water and for multiple days. Did the authors look at single systemic administration of Metformin compared to chronic administration in drinking water? What were the effects of giving Metformin prior to irradiation compared to after irradiation?
Figure 3 is histochemistry showing Claudin3, Villin, and Ki-67 in the intestine as measures of junction performance. These results are provocative and interesting and add to the data in Figures 1 and 2. Quantitative information in panel D is very helpful.
Figure 4 shows the effect of Metformin promoting stem cell integrity with the Wnt/b-catenin pathway activation of Olfm4 and β-catenin. These results are very positive and provocative. Dose response curves of the amount of Metformin given over several different schedules would be helpful, as would a dose response curve of several different radiation doses. It would help to indicate the mouse strain and the gender in the legend to Figure 4.
Figure 5 is a study of organoid formation and studies morphology and percentage of budding organoids. Panel A shows morphology. Panel B shows MTT (This should be explained in the legend and not left as an abbreviation.). Panel C is the proliferation of cells treated with Metformin, and the levels of Myc, Ascl2, and Axin2 in panel E.
This is very exciting and provocative data. A dose response curve of dose of Metformin in single administration, chronic administration, or administration prior to or after irradiation would be very helpful. A dose response curve of different doses of irradiation would be helpful.
Materials and Methods Section
It is clear that C57BL/6J mice were used. Were these male or female? It appears that the animals were given single dose of 13.5 Gy to the abdomen. Need information as to how the thoracic cavity and head and neck were shielded, and hind limbs were shielded should be shown.
500 mg/kg/day of Metformin were given orally for 6 days. Were other schedules tried? Was a single dose given prior to irradiation? A single dose after irradiation? Were other doses of irradiation used?
The section on histology, rt-PCR, and bacterial translocation assay are well described. The study of culture, budding, and MTT assay are shown on page 12. What is MTT? Many readers will not know what that is.
Discussion:
Is there clinical information on the intestinal toxicity to radiotherapy patients, who have been on Metformin? For example, for concomitant Diabetes. It would be an interesting discussion point. The authors may wish to carry out future studies on mice that are defective in Wnt/β-catenin pathway to confirm their findings.
Author Response
#Reviewer 1
Comments and Suggestions for Authors
The authors have reported on the effects of Metformin, a drug with multiple biologic activities and used in the treatment of Diabetes, to radiation protection with respect to enteropathy and the inflammatory response in the intestine. The authors look at multiple parameters of biology, then test them after irradiation. The studies are of interest and are provocative. I have given the authors some suggestions for future studies, and how they can amplify the current observations to a more mechanistic approach to understand the effects of Metformin.
Specific Comments:
Abstract:
The abstract should contain some quantitative information: villus length, number of intestinal stem cells, or other measures of goblet or intestinal barrier function.
- To respond to the comments, we revised the sentence and added the words.
Introduction:
The authors should summarize many of the other radiation protectors and mitigators that have been reported with respect to the gastrointestinal syndrome. Some reference to the animal models that have been used in addition to the mouse. Radiation doses that induce the GI Syndrome. Some explanation of the difference between hematopoietic syndrome and GI syndrome should be included.
- To respond to the comments, we removed and added the sentences.
Results:
The results are largely restricted to quantitative information, such as in Figure 1, which looks at body weight, H & E stain of intestine, histochemistry and myeloperoxidase activity is a neutrophil marker for inflammation, and villus length. This is largely quantitative and histopathology. It would help to have a dose response curve of various doses of irradiation and various times of measurement. Pathologic sections were taken at one time point. The survival curve of animals with each of several different radiation doses should be shown. Body weight is helpful, but the survival would be of interest.
- Thanks for your comments. We agree your comments. To identify the survival effects of metformin in radiation-induced intestinal injury, we previously performed additional experiments. Unfortunately, post-treatment of metformin could not improve the survival rate in 15 Gy abdominal irradiation model. It is the therapeutic limitation of metformin. We plan to the further study that the combined drugs for improving the survival rate in radiation-induced intestinal injury model.
Someone should mention the mouse strain and whether males or females were used.
- To respond to the comments, we added the word in section of result 2.1.
The authors use mucin2 (Muc2) as a marker of differentiated goblet cells. In Figure 2, they show goblet cells as a measure of intestinal barrier function. However, barrier function is usually quantitated by either fluorescent beads gavaged into the intestine, and then measured as leaking into the peripheral blood, or other measures of integrity of tight junctions between epithelial cells such as occludins, or i-CAM, or other adhesion molecules. However, these results are interesting. The authors look at bacterial translocation in panel A. They use Alcian Blue and PAS staining, as well as, Muc2 staining. In the Muc2 staining, the magnification factors are different and significant damage in the irradiation panel that is shown in the middle that is not seen in either the IR+Met or Control. A dose response curve of irradiation and a dose response curve of Metformin should be shown. Metformin is often given in drinking water and for multiple days. Did the authors look at single systemic administration of Metformin compared to chronic administration in drinking water? What were the effects of giving Metformin prior to irradiation compared to after irradiation?
- Our purpose of this study is the mitigator development of metformin for radiation-induced intestinal injury. Therefore, the effects of pre-treatment metformin is the out of scope in this study. To response the reviewer comments, we performed additional experiments. We organized the group
- Control,
- IR 12 Gy,
- Pretreated metformin 100 mg/kg once with IR 12 Gy (M100+IR12Gy),
- Pretreated metformin 500 mg/kg once with IR 12 Gy (M500+IR12Gy),
- IR 12 Gy and post-treatment metformin 100 mg/kg/daily for 6 days (IR12Gy+M100),
- IR 12 Gy and post-treatment metformin 500 mg/kg/daily for 6 days (IR12Gy+M500)
- IR 13.5 Gy,
- Pretreated metformin 100 mg/kg once with IR 13.5 Gy (M100+IR13.5Gy),
- Pretreated metformin 500 mg/kg once with IR 13.5 Gy (M500+IR13.5Gy),
- IR 13.5 Gy and post-treatment metformin 100 mg/kg/daily for 6 days (IR13.5Gy+M100),
- IR 13.5 Gy and post-treatment metformin 500 mg/kg/daily for 6 days (IR13.5Gy+M500)
In 12 Gy irradiated model, the intestinal histological score did not severe levels. Metformin treatment partially protected radiation-induced intestinal damage. Otherwise, post-treatment of metformin did not showed significant difference in histological score, villi length, and crypt counts. In 13.5 Gy irradiation model, we identified the severe histological damage in small intestine. And pre-treatment of metformin showed the protective effects in radiation intestinal damage in M100 and M500 group. we also identified the mitigative effects of metformin in IR 13.5Gy+M100 and IR 13.5Gy+M500 group, and there was no difference between M 100 and M 500 group.
Figure 3 is histochemistry showing Claudin3, Villin, and Ki-67 in the intestine as measures of junction performance. These results are provocative and interesting and add to the data in Figures 1 and 2. Quantitative information in panel D is very helpful.
- Thank you for the comments. In section 2.3, we suggested that metformin enhanced tight junction expression and epithelial proliferation in mouse model. There is a possibility that the main focus may be blurred to add the data of proliferation and junction expression in figure 1 and 2. Therefore I want to remain the section 2.3 as independent section.
Quantitative information in panel D is very helpful.
- To respond to the comments, we add the sentence.
Figure 4 shows the effect of Metformin promoting stem cell integrity with the Wnt/b-catenin pathway activation of Olfm4 and β-catenin. These results are very positive and provocative. Dose response curves of the amount of Metformin given over several different schedules would be helpful, as would a dose response curve of several different radiation doses. It would help to indicate the mouse strain and the gender in the legend to Figure 4.
- To respond to the comments, we conducted additional experiments. In 13.5 Gy irradiated mouse, M100 and M500 treatment did not show significant difference in histological scores, crypt counts, and villi length. Because 12 Gy irradiated mouse model has mild intestinal damage, we estimated maintain of stem cell property in low dose radiation exposure.
Figure 5 is a study of organoid formation and studies morphology and percentage of budding organoids. Panel A shows morphology. Panel B shows MTT (This should be explained in the legend and not left as an abbreviation.). Panel C is the proliferation of cells treated with Metformin, and the levels of Myc, Ascl2, and Axin2 in panel E.
This is very exciting and provocative data. A dose response curve of dose of Metformin in single administration, chronic administration, or administration prior to or after irradiation would be very helpful. A dose response curve of different doses of irradiation would be helpful.
- The aim of this study is the development of radiation mitigator (as metformin) for intestinal damage, not radiation protectors. To respond to the comments, we performed additional experiments. There was mild intestinal damage in 12 Gy irradiated mouse model. And pre- or post- treatment of metformin did not showed significant protective or mitigative effects in low dose irradiation. Otherwise, pre- and post-treatment of metformin in high dose irradiation (13.5Gy) showed a significant protective effects and mitigative effects. Especially, pre-treatment of metformin showed more therapeutic effects in radiation intestinal damage compared with post-treated metformin group. Also, there was no difference between M100 and M500 group. Unfortunately, Metformin treatment did not increased survival rate in 15 Gy abdominal irradiation. Therefore, we will develop the metformin-based combined drugs for improving survival property in radiation-induced intestinal injury as a further study. The additional data was not include in the main figure, because the additional data did not correspond to the subject of this study.
Materials and Methods Section
It is clear that C57BL/6J mice were used. Were these male or female? It appears that the animals were given single dose of 13.5 Gy to the abdomen. Need information as to how the thoracic cavity and head and neck were shielded, and hind limbs were shielded should be shown.
- We mentioned ‘Seven-week-old male C57BL/6 mice, in senction 4.1. We set the irradiation area of mouse abdomen by regulating x-ray beam field size of X-RAD 320 X-ray irradiator (Softex, Gyeonggi-do, Korea).
500 mg/kg/day of Metformin were given orally for 6 days. Were other schedules tried? Was a single dose given prior to irradiation? A single dose after irradiation? Were other doses of irradiation used?
- To respond to the comments, we performed additional experiments. However, the additional data was not include in the main figure, because the additional data did not correspond to the subject of this study.
The section on histology, rt-PCR, and bacterial translocation assay are well described. The study of culture, budding, and MTT assay are shown on page 12. What is MTT? Many readers will not know what that is.
- To respond to the comments, we added the sentence.
Discussion:
Is there clinical information on the intestinal toxicity to radiotherapy patients, who have been on Metformin? For example, for concomitant Diabetes. It would be an interesting discussion point. The authors may wish to carry out future studies on mice that are defective in Wnt/β-catenin pathway to confirm their findings.
- To respond your comments, we added the sentence and reference in section 3.
The authors may wish to carry out future studies on mice that are defective in Wnt/β-catenin pathway to confirm their findings.
- Thank you for the comments. In further study, we study on the effects of metformin dependent on Wnt/b-catenin pathway.

Reviewer 2 Report
The authors describe that Metformin alleviates radiation-induced enteropathy and partially inhibits inflammation response in a mouse model and that Metformin protects intestinal barrier function by indirectly regulating goblet cell maturation in radiation damage. They also demonstrate that Metformin enhances tight junction expression with epithelial proliferation in the irradiated intestine in vivo and vitro. Interesting results are found, however, several revisions are needed.
- The diagram of this in vivo experiment will help readers understand easily on this study.
- In Figure 1, the inflammation score will be needed. The below reference show the scoring system of colitis in mouse. "Development of Reliable, Valid and Responsive Scoring Systems for Endoscopy and Histology in Animal Models for Inflammatory Bowel Disease. J Crohns Colitis. 2018 Jun 28;12(7):794-803. doi: 10.1093/ecco-jcc/jjy035. "
- In Figure 4, mRNA expression levels of stem cell markers are measured. Immunohistochemistry will be needed for Lgr5 and beta-catenin, at least.
- It is better to perform the FITC-dextran test as a functional "leaky gut" test.
Author Response
#Reviewer 2
Open Review
Comments and Suggestions for Authors
The authors describe that Metformin alleviates radiation-induced enteropathy and partially inhibits inflammation response in a mouse model and that Metformin protects intestinal barrier function by indirectly regulating goblet cell maturation in radiation damage. They also demonstrate that Metformin enhances tight junction expression with epithelial proliferation in the irradiated intestine in vivo and vitro. Interesting results are found, however, several revisions are needed.
- The diagram of this in vivo experiment will help readers understand easily on this study.
- Thank you for the comments. To respond to the comments, we added experimental schematics in figure1A.
- In Figure 1, the inflammation score will be needed. The below reference show the scoring system of colitis in mouse. "Development of Reliable, Valid and Responsive Scoring Systems for Endoscopy and Histology in Animal Models for Inflammatory Bowel Disease. J Crohns Colitis. 2018 Jun 28;12(7):794-803. doi: 10.1093/ecco-jcc/jjy035. "
- Thank you for the comments. IBD is characterized with severe inflammation with barrier damage in colon lesion. Therefore crypt damage in IBD is occurred by severe inflammatory response. Otherwise, radiation exposure directly induces apoptosis of proliferating cell in the crypts, and then occurs inflammation reaction in radiation-indued intestinal damage. There is a little difference between IBD and radiation-induced enteropathy. We thought that it would be difficult to directly apply the scoring of this reference commented by the reviewer1, so we reflected the inflammatory infiltration score system. The scoring data showed in figure 1H.
- In Figure 4, mRNA expression levels of stem cell markers are measured. Immunohistochemistry will be needed for Lgr5 and beta-catenin, at least.
- Figure 4B showed the b-catenin expression in the small intestine. As your comments, we think it is very important to check the expression of lgr5 for stem cell property. We also tried to identify LGR5 protein expression using Lgr5 antibodies from several companies, but there were unsuccessful. In Figure 4, stem cell property was confirmed using Olfm4 marker instead of Lgr5.
- It is better to perform the FITC-dextran test as a functional "leaky gut" test.
- FITC-dextran assay are applied to identify the intestinal barrier function. In several research, we confirmed that bacterial translocation in mesenteric lymph node also is a good experimental method for examining the barrier function in radiation-induced intestinal injury (Shim et al., 2017 Toxicology and Applied Pharmacology; Jang et al., 2019 Frontiers in Pharmacology; Kwak et al., 2021 Biomedicine). In addition, while the FITC-dextran assay identify intestinal permeability, bacterial translocation assay is an experimental method that can confirm the movement of bacteria into internal organs that can induce systemic sepsis along with permeability.

Round 2
Reviewer 2 Report
I am satisfied. Good job.
Author Response
Thank you very much.